# Combining Deep Learning and Graph-Theoretic Brain Features to Detect Posttraumatic Stress Disorder at the Individual Level

**DOI:** 10.3390/diagnostics11081416

**Published:** 2021-08-05

**Authors:** Ziyu Zhu, Du Lei, Kun Qin, Xueling Suo, Wenbin Li, Lingjiang Li, Melissa P. DelBello, John A. Sweeney, Qiyong Gong

**Affiliations:** 1Huaxi MR Research Center (HMRRC), Department of Radiology, West China Hospital of Sichuan University, Chengdu 610041, China; zzy31600@foxmail.com (Z.Z.); qk_means0902@foxmail.com (K.Q.); xuelingsuo_hmrrc@163.com (X.S.); liwenbinchn@foxmail.com (W.L.); sweenej5@ucmail.uc.edu (J.A.S.); 2Department of Psychiatry and Behavioral Neuroscience, University of Cincinnati, Cincinnati, OH 45219, USA; leidu@ucmail.uc.edu (D.L.); DELBELMP@UCMAIL.UC.EDU (M.P.D.); 3Mental Health Institute, The Second Xiangya Hospital of Central South University, Changsha 410008, China; llj2920@163.com; 4Research Unit of Psychoradiology, Chinese Academy of Medical Sciences, Chengdu 610000, China; 5Functional and Molecular Imaging Key Laboratory of Sichuan Province, Chengdu 610000, China

**Keywords:** graph theory, posttraumatic stress disorder, deep learning, support vector machine, salience network, neuroimaging, psychoradiology

## Abstract

Previous studies using resting-state functional MRI (rs-fMRI) have revealed alterations in graphical metrics in groups of individuals with posttraumatic stress disorder (PTSD). To explore the ability of graph measures to diagnose PTSD and capture its essential features in individual patients, we used a deep learning (DL) model based on a graph-theoretic approach to discriminate PTSD from trauma-exposed non-PTSD at the individual level and to identify its most discriminant features. Our study was performed on rs-fMRI data from 91 individuals with PTSD and 126 trauma-exposed non-PTSD patients. To evaluate our DL method, we used the traditional support vector machine (SVM) classifier as a reference. Our results showed that the proposed DL model allowed single-subject discrimination of PTSD and trauma-exposed non-PTSD individuals with higher accuracy (average: 80%) than the traditional SVM (average: 57.7%). The top 10 DL features were identified within the default mode, central executive, and salience networks; the first two of these networks were also identified in the SVM classification. We also found that nodal efficiency in the left fusiform gyrus was negatively correlated with the Clinician Administered PTSD Scale score. These findings demonstrate that DL based on graphical features is a promising method for assisting in the diagnosis of PTSD.

## 1. Introduction

Exposure to a disaster has been associated with a variety of mental health consequences [1]. Prior research has reported that survivors of natural disasters are highly likely to develop posttraumatic stress disorder (PTSD) [2], which is characterized by a heightened sensitivity to potential threats (including those related to the initial traumatic experience) and can be devastating to the affected individuals and their families. Many survivors exhibit posttraumatic stress symptoms in the weeks and months after exposure [3], but waiting for individuals to develop PTSD before intervening can delay preventive or early effective treatment. Besides, chronic PTSD is associated with a host of physical ailments (e.g., irritable bowel syndrome [4]). It can be particularly pernicious and disabling for many across the lifespan. There is therefore an urgent need to find an accurate method to diagnose PTSD as early as possible after major acute stress.

The current diagnostic criteria for PTSD rely on clinical interviews. Direct examination of brain function patterns provides an alternative/complementary approach. With advances in neuroimaging techniques (i.e., psychoradiology [5]), an increasing number of studies using various imaging modalities have consistently found the brain to be a large, interacting, complex network with nontrivial topological properties, the so-called small-world architecture [6,7,8]. This is an attractive model for assessing the connectivity of the nervous system because the combination of highly connected hubs and short path length confers the capability for both specialized and modular processing in local neighborhoods in a distributed or integrated manner [9,10].

In our previous study [11] on the topological organization of the functional connectome of individuals with PTSD, we demonstrated that topological alterations predominantly involved the default-mode network (DMN) and the salience network (SN), which are associated with affective processing [12] and interoceptive-autonomic processing [13], respectively. Marked differences in network topology have also been found in various brain diseases, such as traumatic brain injury [14], Alzheimer’s disease [15], autism spectrum disorder [16], schizophrenia [17], major depressive disorder [18], and attention-deficit/hyperactivity disorder [19] and may underlie the pathogenesis of these disorders. To better analyze such complex networks, the application of graph theory, which quantitatively examines all possible network connections and elucidates key topological properties of the overall network and subnetworks and the function of regions within local and global networks, has been increasingly and extensively applied [20]. Further, methods have been developed to apply this approach for individual patients rather than only for group data.

Advancements in deep learning (DL) have shown promising outcomes for prediction and characterization for individuals with brain-based disorders [21]. This set of techniques infers hierarchical feature representations from the lowest level and then gradually establishes more complex representations from the previous level, enabling the inference of complex nonlinear relationships [22]. Compared with traditional machine learning algorithms (e.g., support vector machine (SVM) [23]), DL may yield higher accuracy in disease classification and can automatically extract the optimal representations from the raw data without the preselection of features [21,24]. To date, a number of studies have applied DL techniques to the classification of psychiatric disorders based on anatomical brain images obtained through MRI (e.g., [25]) or functional brain images [26] or by combining structural and functional neuroimaging data (e.g., [27,28]). However, no researchers have used topological properties as inputs in the classification of PTSD.

The purpose of this study was to extract the topology of the functional brain connectome in individuals with PTSD and find a suitable classification model with high accuracy by training the datasets. We also made efforts to identify the features contributing most to single-subject classification and their association with symptom severity. In light of the increasing understanding of PTSD as the manifestation of abnormalities in topological properties [11,14,29,30], we hypothesized that: (i) the application of DL to graph-based analytic metrics would allow the accurate identification of PTSD in trauma-exposed individuals; specifically, that it would provide higher accuracy than using the traditional SVM model; and (ii) based on previous findings [11,31], the aberrant interaction between DMN, central executive network (CEN) and SN may underlie the pathophysiology of PTSD.

## 2. Materials and Methods

### 2.1. Participants

The participants were recruited after the Wenchuan earthquake on 12 May 2008, considered one of the most devastating natural disasters in Chinese history [32]. The acquisition of neuroimaging and clinical data from survivors took place between 10 and 15 months after the event. All recruited participants witnessed and physically experienced life-threatening or other traumatic events related to this horrific earthquake. After a complete description of the study, the participants signed informed consent forms. The study was approved by the Research Ethics Committee of West China Hospital of Sichuan University.

Participants were included if they met the following criteria: (i) age between 18 and 65 years, (ii) right-handed, (iii) no psychiatric history or PTSD prior to the earthquake, (iv) an intelligence quotient >80, (vi) no brain traumatic history or any neurological disease, and (vii) no contraindication to MR imaging. Those who met the inclusion criteria were assessed using the PTSD Checklist (PCL) [33] and the Clinician Administered PTSD Scale (CAPS) [34]. The subjects were eligible for inclusion in the PTSD group when the PCL score was ≥35 points and the CAPS score was ≥50 points [11]. An age- and sex-matched trauma-exposed non-PTSD group was formed from those with a PCL score < 30 points and a CAPS score < 35 points [11,35]. All subjects were further evaluated by experienced psychiatrists (with 30 years of experience) to confirm the PTSD diagnosis and exclude psychiatric comorbidities using the Structured Clinical Interview for the Diagnostic and Statistical Manual of Mental Disorders, Fourth Edition (DSM-IV) [36]. Exclusion criteria included: (i) the presence of any psychiatric comorbidities; (ii) the lack of early life trauma exposure or alcohol abuse; and (iii) treatment with psychiatric medications within two months before recruitment for the MRI scan. The checklists and scales were administered in Mandarin. Moreover, after the MR scan, data with excessive head movements (translation > 1.0 mm, rotation > 1°), brain lesions or obvious artifacts were discarded. Two authors (Z.Z. and D.L.) checked the data to establish eligibility of the MRI data for inclusion; any disagreement was mediated, and a consensus was reached. The demographic and clinical data were analyzed using SPSS version 16.0. All tests were two tailed.

### 2.2. Image Acquisition

Brain scans were performed using a Signa EXCITE 3.0T MRI system (GE EXCITE, Milwaukee, WI) with an eight-channel phased-array head coil. During the resting-state functional MRI (rs-fMRI) examination, subjects were requested to keep their eyes closed, relax and let their minds wander. Sequence parameters were as follows: repetition time/echo time (TR/TE) = 2000/30 ms; flip angle = 90°; number of axial sections per volume = 30; section thickness = 5 mm (no section gap); matrix = 64 × 64; field of view (FOV) = 240 × 240 mm^2^; voxel size = 3.75 × 3.75 × 5 mm^3^. Each functional run generated 200 image volumes, resulting in a total scan time of 400 s.

### 2.3. Data Preprocessing

Preprocessing of the functional images was carried out using Statistical Parametric Mapping 12 (SPM12) (http://www.fil.ion.ucl.ac.uk/spm/; accessed on 22 November 2020), which is a recent and advanced standard neuroimaging processing software. The preprocessing pipeline, developed according to the literature [11,37,38], is shown in Figure 1.

First, the initial 10 time points were discarded to remove the impact of magnetization stabilization. Then, slice-timing adjustment and realignment for head motion were performed on the remaining images to reduce intravolume acquisition time mismatch and intervolume spatial displacement, respectively. Since the anatomy and size of the brain among the participants may be different, we normalized the corrected images to the standard Montreal Neurological Institute (MNI) atlas at a 3 × 3 × 3 mm^3^ resolution. Next, a Gaussian filter with a full-width at half-maximum (FWHM) of 4 mm was used to smooth the normalized data, and the smoothed data were detrended and subsequently passed through a bandpass filter (0.01–0.08 Hz) to remove low-frequency drift and high-frequency physiological noise. Finally, covariate regressions were performed to eliminate the influence of nuisance covariates (white matter signal, cerebrospinal fluid signal, and motion parameters) on the results.

### 2.4. Network Construction

A network is a mathematical representation of a real-world complex system and is composed of nodes (vertices) and edges (links) between pairs of nodes. We constructed the network by using the Gretna toolbox (http://www.nitrc.org/projects/gretna/; accessed on 28 January 2021), which allows researchers to perform comprehensive analysis on the topology of brain connectome. First, we divided the cerebrum into 90 regions of interest (ROIs) according to the Automated Anatomical Labeling (AAL) template [39], each of which was defined as a node. For each subject, the representative time series of each region was obtained by averaging the time series of all voxels in the region. Then, we computed the partial correlations between all pairs of nodes, which can attenuate the contribution of other sources of covariance and be estimated as the edges of the network. A partial correlation matrix (*n* × *n*, where *n* is the number of brain regions, here set to 90) was ultimately obtained and converted into a binary matrix. To remove spurious functional edges, we applied a wide range of sparsity (S) thresholds (defined as the total number of edges in a graph divided by the maximum possible number of edges; here ranging from 0.10 to 0.34 in steps of 0.01) [11] to all correlation matrices rather than a single threshold. To provide a summarizing scalar value for the selected threshold space, we calculated the area under the curve (AUC) for each network metric to characterize the topological organization of brain features [40,41].

The network (graph) was represented by the binarized matrix calculated above. For each of the brain networks at each sparsity threshold, we calculated seven global metrics and three nodal centrality metrics. Global metrics, which include five small-world parameters (characteristic path length Lp, clustering coefficient Cp, normalized clustering coefficient γ, normalized characteristic path length λ and small-worldness σ), and two network efficiency parameters (global efficiency Eglob, local efficiency Eloc), reflect the whole network topological architecture [42]. Nodal centrality metrics involving degree, efficiency and betweenness reflect the regional topological centralities [43]. We thus obtained a 277-dimensional graphic feature vector, in which the first seven features were the global metrics, and the rest were nodal centrality metrics, for the 90 AAL regions.

### 2.5. Machine Learning Model

We applied a two-stage classification pipeline DL model to differentiate PTSD from trauma-exposed non-PTSD [22]. One strength of this approach is that the neural network can produce a higher-order (nonlinear) representation of the features. Such models have been reported to outperform traditional machine learning and feature engineering methods in an application to predicting future autism development in at-risk infants [22]. This prediction pipeline includes a DL-based dimensionality reduction stage followed by a SVM classification stage. The analyses of these two stages were programmed using Python, in which the neural network was implemented in the Pytorch library [44].

The initial stage of the pipeline is DL-based dimensionality reduction, which can be further divided into two steps: (i) a binary operator and (ii) a deep learning network. First, we applied a trained binary masking operator to transform the real-valued vector into a binary feature vector. To ensure independence between the training set and the verification set, the binary threshold was estimated from the training set and applied to the verification set in each cross-validation cycle. Second, we trained a deep network using the high-dimension binary feature vectors. The second stage is the classification of extracted features with an SVM. The SVM method was implemented in LIBSVM (http://www.csie.ntu.edu.tw/~cjlin/libsvm/; accessed on 17 April 2021), which can be used to construct classification models based on graph metrics. Low dimension codes and the binary training labels were input into a linear SVM classifier. During the training process at the SVM stage, we performed five-fold nested cross-validation to find the optimal hyperparameter via grid search. After the training, ten-fold stratified cross-validation was used to assess the reliability of the classification model. Details of the prediction pipeline have been described elsewhere [22].

To estimate the significance for our classification model, a nonparametric permutation test was conducted to calculate the *p* value for the balanced accuracy. This randomization procedure was repeated 1000 times with a different random permutation of the training group labels. Then, we calculated the number of times that the balance accuracy of the permuted labels was higher than that of the real labels and divided this number by 1000 to calculate the *p* value.

Furthermore, we used a traditional SVM classifier as a reference to evaluate our deep learning method. We trained this traditional classifier on the same data and evaluated its classification performance. No feature selection was performed to maintain the same input data from the deep learning approach. The measure used during the grid search was obtained from the validation sets using the same 10-fold cross-validation process and the same training sets as the DL optimization process. This comparison helped us verify which type of approach was the better discriminator between PTSD and trauma-exposed non-PTSD.

### 2.6. Identification of Features with the Greatest Contribution

One of the main advantages of DL is the ability to automatically identify the most useful features for classification from the raw data without requiring prior feature selection. We extracted all the weight matrices in the fully trained DL network {$\bar{W}$_1_, $\bar{W}$_2_, $\bar{W}$_3_}, where the $\bar{W}$_*i*_ matrix denotes the weight matrix connecting two adjacent layers in {*l*_1_, *l*_2_, *l*_3_, *l*_4_}. The contribution of each node in *l*_*i*_ was estimated from the weight matrix $\bar{W}$_*i*_ (details can be found in [22]). We started from *l*_4_ and worked backwards through the median DL network, estimating the contribution of each node in *l*_3_, then eliminated nodes with small contributions as much as possible. The partition of weight matrix $\bar{W}$_3_ was used to estimate the contribution of the remaining nodes in *l*_3_; the nodes whose summed contributions represented greater than 50% of the weight contribution of *l*_4_ were kept. This calculation was propagated backward until we reached the input layer, where the contributions of the raw features are available. Finally, the top 10 features contributing most to the DL and SVM models were reported.

### 2.7. Correlation Analysis

Correlation analysis was performed between the top 10 features contributing most to the two classification models and the CAPS scores of the individuals with PTSD. A nominal significance level of *p* < 0.05 was used for these exploratory analyses, and a false discovery rate (FDR)-corrected significance level of *p* < 0.05 was used for regional graph metrics to correct for multiple comparisons.

## 3. Results

### 3.1. Demographic and Clinical Characteristics

Table 1 summarizes the sociodemographic and clinical features of the study participants. There was no significant difference in age, sex, or education between PTSD and trauma-exposed non-PTSD (*p* > 0.05).

### 3.2. Classification Performance

The single-subject classification of PTSD and trauma-exposed non-PTSD using graph-based topological metrics was assessed for accuracy, sensitivity, and specificity. The two-stage prediction pipeline approach showed better classification performance (average accuracy of classification: 80.0%, average sensitivity: 80.9%; average specificity: 79.2%; *p* < 0.001) than the traditional SVM approach (average accuracy of classification: 57.7%, average sensitivity: 53.2%; average specificity: 62.2%; *p* < 0.001).

### 3.3. Regions with the Greatest Contribution to Single-Subject Classification

To identify the classification pattern in individual with PTSD and trauma-exposed non-PTSD group, we proceeded to investigated feature contributions to the DL model and the SVM model. The 10 features with the highest mean values across the two models are reported in Table 2 and represented graphically in Figure 2. It can be seen that CEN (including the triangular part of inferior frontal gyrus, middle frontal gyrus), DMN (including the angular gyrus and the superior temporal gyrus), and SN (including the putamen) were the main regions contributing to DL classification performance. In contrast, the CEN (including the triangular part of inferior frontal gyrus and orbital part of middle frontal gyrus) and DMN (including the orbital part of superior frontal gyrus and the middle temporal gyrus) provided the greatest contribution to the classification performance of the SVM.

### 3.4. Relationship between Topological Metrics and Clinical Variables

Only the graphical topological property of the fusiform gyrus obtained by the DL model was found to be significantly correlated with CAPS scores (*p* = 0.038) (Figure 3). However, it did not survive FDR correction (*p* > 0.05). None of the top 10 SVM features showed a significant correlation with CAPS scores.

## 4. Discussion

This study examined the altered graphic topological features extracted from resting-state functional neuroimaging data in individuals with PTSD. We demonstrated that these features could be applied to distinguish individuals with PTSD from trauma-exposed non-PTSD individuals using a two-stage classification pipeline.

In keeping with our first hypothesis, the application of DL (the two-stage prediction pipeline) to graph-based analytic metrics was found to be a powerful tool for differentiating PTSD from trauma-exposed non-PTSD at the level of the individual and achieved higher accuracy than the SVM approach. We extended the preliminary ideas in Hazlett et al. [22] and provided a set of experiments to evaluate the proposed modeling. Our method achieved promising classification results, which may be explained as follows. First, we combined a graph-theoretic approach with an advanced deep learning method. Graph theory can effectively describe different aspects of the brain network; specifically, it examines all possible network connections and elucidates key topological properties of the overall network and subnetworks and the function of regions within local and global networks [45]. Thus, it allows for an increasingly sophisticated analysis of brain networks at a level of complexity relative to previous studies evaluating region-by-region functional and structural brain features. 

Additionally, the powerful DL method was able to learn subtle hidden patterns from high-dimensional neuroimaging data and automatically extract optimal features from the raw data through consecutive nonlinear transformations, ensuring that the learned features were the most discriminating between the two populations [21,26]. Second, the DL model we used contains a two-stage classification pipeline, including a DL-based dimensionality reduction stage followed by an SVM classification stage. It has been demonstrated that this model has better classification performance than other classification methods, such as sparse learning + SVM, deep classification only, and two proposed principal component analysis + SVM classification methods [22]. Our findings are consistent with these observations. Previous studies using structural or functional brain imaging data have implemented a variety of methods to classify and predict PTSD, with the accuracy of ML methods ranging from 67% to 94.2% [46,47,48,49,50]. The average diagnostic accuracy of the methods used in our study was not exceptionally high; one possible explanation is the ambiguity in fitting the model to the target mapping due to the limited amount of training data. An alternative explanation is that while the extraction of graphical topological properties from the whole brain time series can reveal important functional features at the group level, some important information about network-level functioning at the individual level may be lost during the computation. Also, we compared PTSD patients with individuals who also had experienced acute stress, a more clinically relevant differentiation and likely a more challenging one, rather than community controls. 

Our current study highlights the potential of graphical topological properties of resting-state fMRI data in characterizing brain diseases at the individual level. This notion is supported by several previous studies that applied graphical topological properties to SVM models. For example, prior studies have used this approach for successfully classifying individuals with major depressive disorder [18], schizophrenia [51], and Alzheimer’s disease [52] from healthy controls, with accuracy of 79.27%, 95.00%, and 71.95% respectively.

The characteristic path length Lp was the most discriminative graphic property for PTSD classification with the DL approach. In contrast, there was no significant contribution from any of the global metrics to classification when using the SVM approach. The characteristic path length Lp is an emergent property of the graph, defined as the average shortest path connecting any 2 vertices on the graph. It is a global metric that corresponds to a basic principle of brain functional organization, namely, functional integration [53]. Thus, our results indicated that the DL classification was driven by both global and nodal measures, while the SVM classification was primarily driven by nodal measures. This notion is consistent with Algunaid’s study [51], in which an SVM classifier based on a graph-theoretic approach was used. That study indicated that the local graph measures outperformed the global graph measures in distinguishing between control subjects and individuals with schizophrenia. 

Notably, previous studies have reported brain structural and function changes in PTSD relative to either trauma-exposed controls without PTSD or non-traumatized healthy controls [54,55,56]. It is difficult to conclude whether the observed alterations were related to disease or traumatic stress. Therefore, for controls, we selected a population that was exposed to the earthquake without developing PTSD to control variables. However, further studies are desirable with an additional group of non-traumatized individuals to provide more comprehensive insight into the functional networks underlying PTSD rather than life stress more generally. Furthermore, increasing studies have reported brain or spine injury after traumatic event [57,58,59]. Although we excluded the individuals with brain traumatic history to minimize the impact of confounding factors, future studies may include this population.

The present study has several limitations. First, the brain parcellation template we selected for constructing the brain networks may have affected the network analysis results [40]. Future studies might verify our results using newer brain atlases, such as the Power 264-region atlas [60] and the Dosenbach’s 160 functional atlas [61]. Second, a lack of sufficiently large samples at individual sites may lead to poor generalizability in the automatic diagnostic classification of heterogeneous psychiatric disorders. Larger population-based rs-fMRI databases may help to evaluate the variability and stability of large-scale networks in the general population. Third, as the fear circuitry and dysphoric PTSD symptoms may emerge to different extents over time, the possible neuroanatomical changes during the development of PTSD may also be taken into account. Finally, several studies have accurately predicted or classified PTSD using a broad array of theory-driven cognitive and neurobiological factors [62,63,64]. Future studies combining imaging data with clinical and other biological data might help train more robust DL models. More work is needed by way of replication and adding non-imaging features into classifiers to establish the method that is optimal for clinical application.

## 5. Conclusions

In conclusion, despite the above limitations, we successfully discriminated individuals with PTSD and trauma-exposed non-PTSD and identified the brain regions affected by PTSD using a two-stage graph-theoretic DL approach on resting-state fMRI data. Our proposed method discriminated between PTSD and trauma-exposed non-PTSD by using informative sets of brain graph measures with promising accuracy. The pattern of results suggests that the application of the two-stage prediction pipeline approach would help in the development of more accurate ML algorithms and possibly allow diagnostic classification with higher clinical application value. This may be the first step to building neuroimaging-based discriminative models to predict the onset of PTSD in a high-risk sample or to differentiate PTSD from other disorders with clinical overlap. It is worth noting that the included subjects were untreated and presented with no psychiatric comorbidities. It excludes the effect of the drug and co-morbidities, which may make our results more useful for biological mechanistic understanding, but perhaps less representative for wide clinical application. In addition, the regions with the greatest contribution to PTSD classification are key nodes in three major intrinsic connectivity networks (i.e., CEN, DMN, and SN), implying that investigating these three networks may serve as a better representation of the heterogeneous clinical profiles of those individuals with PTSD. Additional validation and extension steps will be needed to assess the clinical applicability of our method.

## Figures and Tables

**Figure 1 diagnostics-11-01416-f001:**
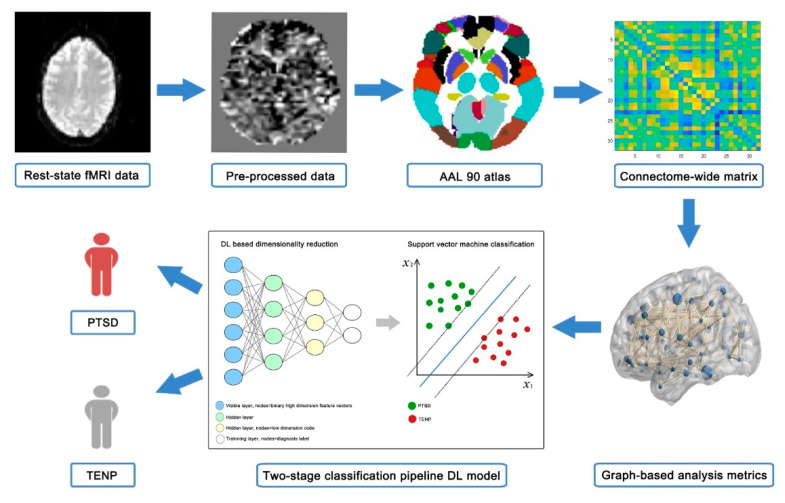
Overview of the employed classification approach showing the main steps of the pipeline. Abbreviation: AAL, automated anatomical labeling; SVM, support vector machine; DL, deep learning; PTSD, posttraumatic stress disorder; TENP, trauma-exposed non-PTSD patients.

**Figure 2 diagnostics-11-01416-f002:**
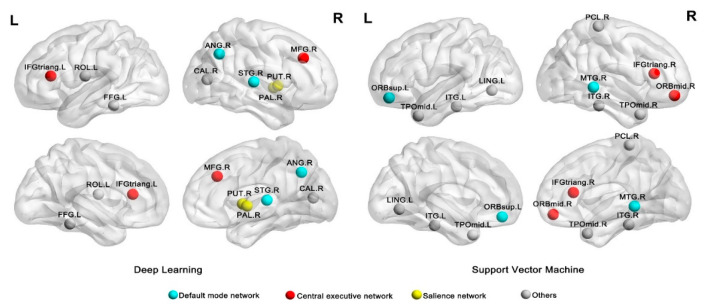
Regions providing the greatest contribution to single-subject classification of patients and controls. The nodes were mapped onto the cortical surfaces by using the BrainNet Viewer package (http://www.nitrc.org/projects/bnv/; accessed on 23 April 2021). Circles represent AAL nodes, blue represents the default mode network, red represents the central executive network, yellow represents the salience network and grey represents others. Abbreviation: Left: IFGtriang, Inferior frontal gyrus, triangular part; PUT, Putamen; ANG, Angular gyrus; STG, Superior temporal gyrus; ROL, Rolandic operculum; CAL, Calcarine fissure and surrounding cortex; FFG, Fusiform gyrus; PAL, pallidum; MFG, Middle frontal gyrus L, left; R, right. Right: ITG, Inferior temporal gyrus; LNG, Lingual gyrus; TPOmid, Temporal pole: middle temporal gyrus; IFGtriang, Inferior frontal gyrus, triangular part; PCL, Paracentral lobule; ORBsup, Superior frontal gyrus, orbital part; MTG, Middle temporal gyrus; ORBmid, Middle frontal gyrus, orbital part; L, left; R, right.

**Figure 3 diagnostics-11-01416-f003:**
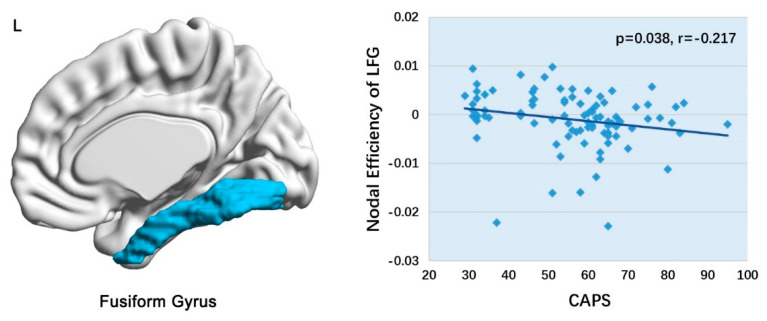
Scatter plot of nodal efficiency of left fusiform gyrus relation to CAPS scores in PTSD. Abbreviation: PTSD, posttraumatic stress disorder; CAPS, clinician-administered PTSD scale; LFG, left fusiform gyrus.

**Table 1 diagnostics-11-01416-t001:** Demographic and clinical characteristics of the participants ^a^.

Variables	PTSD	TENP	*p* Value
Sample size	91	126	-
Age (years) ^b^	42.4 ± 10.2	43.1 ± 9.6	*p* = 0.58 ^c^
Gender (male/female)	29/62	40/86	*p* = 0.985 ^d^
Handedness (right/left)	91/0	126/0	-
Education (years)	7.1 ± 3.0	7.9 ± 3.8	*p* = 0.09 ^c^
PCL	47.7 ± 12.3	28.2 ± 6.0	*p* < 0.001 ^c^
CAPS	56.1 ± 14.9	22.8 ± 8.7	*p* < 0.001 ^c^

**^a^** Data are presented as mean ± standard deviation. ^b^ Age defined at the time of MRI scanning. ^c^ *p* by two-tailed two-sample *t* test. ^d^ *p* by two-tailed Pearson Chi-square test. Abbreviation: PTSD, Posttraumatic Stress Disorder; TENP, trauma-exposed non-PTSD; PCL, PTSD Checklist; CAPS, Clinician-Administered PTSD Scale.

**Table 2 diagnostics-11-01416-t002:** Top 10 most relevant topological properties of brain regions for Deep Learning versus Support Vector Machine classification analysis *.

DL		SVM	
Topological Property	Brain Regions	Topological Property	Brain Regions
Characteristic path length	-	Nodal degree	Inferior temporal gyrus, L
Nodal betweenness	Inferior frontal gyrus, triangular part, L (CEN)	Nodal betweenness	Lingual gyrus, L
Nodal betweenness	Lenticular nucleus, putamen, R (SN)	Nodal degree	Inferior temporal gyrus, R
Nodal betweenness	Angular gyrus, R (DMN)	Nodal degree	Temporal pole: middle temporal gyrus, L
Nodal efficiency	Superior temporal gyrus, R (DMN)	Nodal betweenness	Inferior frontal gyrus, triangular part, R (CEN)
Nodal betweenness	Rolandic operculum, L	Nodal betweenness	Paracentral lobule, R
Nodal betweenness	Calcarine fissure and surrounding cortex, R	Nodal degree	Temporal pole: middle temporal gyrus, R
Nodal efficiency	Fusiform gyrus, L	Nodal betweenness	Superior frontal gyrus, orbital part, L (DMN)
Nodal betweenness	Lenticular nucleus, pallidum, R (SN)	Nodal degree	Middle temporal gyrus, R (DMN)
Nodal betweenness	Middle frontal gyrus, R (CEN)	Nodal betweenness	Middle frontal gyrus, orbital part, R (CEN)

* All brain regions are from AAL (automated anatomical labelling). Abbreviations: CEN, central executive network; DMN, default mode network; SN, salience network; DL, deep learning; SVM, support vector machine; L, left; R, right.

## Data Availability

The data presented in this study are available on request from the corresponding author. Some human data are not publicly available due to data protection reasons.

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
