# Peer review of "Combining Deep Learning and Graph-Theoretic Brain Features to Detect Posttraumatic Stress Disorder at the Individual Level"

_diagnostics, 2021, doi:10.3390/diagnostics11081416_

Round 1

Reviewer 1 Report

I read this report with some interest. I commend the authors on their work, which represents a curious approach towards the early detection of PTSD.

Specific comments:

  1. "... or early effective treatment" - can add that PTSD is also associated with a host of physical ailments, e.g. irritable bowel syndrome (citation: pubmed.ncbi.nlm.nih.gov/30144372). Chronic PTSD can be particularly pernicious and disabling for many across the lifespan.
  2. Were the checklist and scales administered in Mandarin? Please specify.
  3. Did authors correct for multiple comparisons (FDR correction)?
  4. As time elapses after trauma, fear circuitry and dysphoric PTSD symptoms may emerge to different extents or degrees. This is a potential study limitation as well.
  5. Further studies should also be conducted with an additional group of non-traumatized individuals.

Author Response

Comments to the Author 

I read this report with some interest. I commend the authors on their work, which represents a curious approach towards the early detection of PTSD.

Authors Reply: We thank the reviewer for these positive comments. 

  1. "... or early effective treatment" - can add that PTSD is also associated with a host of physical ailments, e.g. irritable bowel syndrome (citation: pubmed.ncbi.nlm.nih.gov/30144372). Chronic PTSD can be particularly pernicious and disabling for many across the lifespan.

Authors Reply: Thank you for this suggestion, which we have implemented in the revised manuscript: “... or early effective treatment. Besides, chronic PTSD is associated with a host of physical ailments (e.g., irritable bowel syndrome [4]). It can be particularly pernicious and disabling for many across the lifespan. ”. 

  1. Were the checklist and scales administered in Mandarin? Please specify.

Authors’ Reply: We thank the reviewer for this suggestion. In the revised manuscript, we have added this point in Materials and Methods section: “The checklists and scales were administered in Mandarin.”.

  1. Did authors correct for multiple comparisons (FDR correction)?

Authors’ Reply: Yes. We adopted FDR correction to address the problem of multiple comparisons, which was mentioned in Materials and Methods.  

  1. As time elapses after trauma, fear circuitry and dysphoric PTSD symptoms may emerge to different extents or degrees. This is a potential study limitation as well.

Authors’ Reply: Thank you for this excellent suggestion. In the revised manuscript, we have added this limitation in Discussion:

“Third, as the fear circuitry and dysphoric PTSD symptoms may emerge to different extents over time, the possible neuroanatomical changes during the development of PTSD may also be taken into account.”.

  1. Further studies should also be conducted with an additional group of non-traumatized individuals.

Authors’ Reply: We thank the reviewer for this suggestion. In the revised manuscript, we have discussed this point in Discussion:

“Notably, previous studies have reported brain structural and function changes in PTSD relative to either trauma-exposed controls without PTSD or non-traumatized healthy controls [53-55]. It is difficult to conclude whether the observed alterations were related to disease or traumatic stress. Therefore, for controls, we selected a population that was exposed to the earthquake without developing PTSD to control variables. However, further studies are desirable with an additional group of non-traumatized individuals to provide more comprehensive insight into the functional networks underlying PTSD rather than life stress more generally.”.

Reviewer 2 Report

This topic is very intersting, please look at these points:

  1. Lines 105: " an intelligence quotient > 80" Can the authors explain the reason for this cutoff?
  2. Lines 179-221 "2.5. Machine learning model": This part is too long and dispersive, please try to merge and reduce it.
  3. Lines 103-121: "2.1. Participants " what about TENP population, can the authors reported its inclusion criteria?
  4. Line 136: The lower part of Figure 1 is not easy to read. Please improve.
  5. Lines 300-3010: About trauma-exposed non-PTSD please look at these 3 important refs: A management algorithm for patients with intracranial pressure monitoring: the Seattle International Severe Traumatic Brain Injury Consensus Conference (SIBICC). Intensive Care Med. 2019 Dec;45(12):1783-1794. doi: 10.1007/s00134-019-05805 ----   Regional and experiential differences in surgeon preference for the treatment of cervical facet injuries: a case study survey with the AO Spine Cervical Classification Validation Group. Eur Spine J. 2021 Feb;30(2):517-523. doi: 10.1007/s00586-020-06535-z.  ---   Head injuries and the risk of concurrent cervical spine fractures. Acta Neurochir (Wien). 2017 May;159(5):907-914. doi: 10.1007/s00701-111-
  6. Lines 358-361: "Third, for controls, we selected a population 358
    that was exposed to the earthquake without.. life stress more generally." Please discuss this point also in the discussion section.

Author Response

Comments to the Author

This topic is very interesting.

Authors Reply: We thank the reviewer for these positive comments.

Please look at these points:

  1. "an intelligence quotient > 80" Can the authors explain the reason for this cutoff?

Authors Reply: According to the current Wechsler Adult Intelligence Scale—Fourth Edition (WAIS–IV) (ref: Urbina, Susana (2011). "Chapter 2: Tests of Intelligence". In Sternberg, Robert J.; Kaufman, Scott Barry (eds.). The Cambridge Handbook of Intelligence. Cambridge: Cambridge University Press. pp. 20–38. ISBN 978-0-521-73911-5.), the average IQ scores of normal adults range from 80 to 119. We used this inclusion criterion to minimize the impact of cognitive impairment on the results.

  1. "5. Machine learning model": This part is too long and dispersive, please try to merge and reduce it.

Authors’ Reply: We thank the reviewer for this suggestion. In the revised manuscript, we have rewritten the part "2.5. Machine learning model". (Page 5)

  1. "2.1. Participants " what about TENP population, can the authors reported its inclusion criteria?

Authors’ Reply: Our inclusion criteria “(i) age between 18 and 65 years, (ii) right-handed, (iii) no psychiatric history or PTSD prior to the earthquake, (iv) an intelligence quotient > 80, (vi) no brain traumatic history or any neurological disease, and (vii) no contraindication to MR imaging” were for both TENP and PTSD population. Additional inclusion criteria for TENP population were also listed in Materials and Methods - “An age- and sex-matched trauma-exposed non-PTSD group was formed from those with a PCL score < 30 points and a CAPS score < 35 points”.

  1. The lower part of Figure 1 is not easy to read. Please improve.

Authors’ Reply: We thank the reviewer for this comment. In the revised manuscript, we have removed the dispensable formulas and simplified the content in Figure 1, which may help readers understand the main steps of the pipeline more easily. (Page 4)

  1. About trauma-exposed non-PTSD please look at these 3 important refs: A management algorithm for patients with intracranial pressure monitoring: the Seattle International Severe Traumatic Brain Injury Consensus Conference (SIBICC). Intensive Care Med. 2019 Dec;45(12):1783-1794. doi: 10.1007/s00134-019-05805 ----   Regional and experiential differences in surgeon preference for the treatment of cervical facet injuries: a case study survey with the AO Spine Cervical Classification Validation Group. Eur Spine J. 2021 Feb;30(2):517-523. doi: 10.1007/s00586-020-06535-z.  ---   Head injuries and the risk of concurrent cervical spine fractures. Acta Neurochir (Wien). 2017 May;159(5):907-914. doi: 10.1007/s00701-111-

Authors’ Reply: We have included these references in our Discussion section:

“Furthermore, increasing studies have reported brain or spine injury after traumatic event [55-57]. Although we excluded the individuals with brain traumatic history to minimize the impact of confounding factors, future studies may include this population.”.

  1. "Third, for controls, we selected a population that was exposed to the earthquake without.. life stress more generally." Please discuss this point also in the discussion section.

Authors’ Reply: We thank the reviewer for this comment. In the revised manuscript, we have added one paragraph to discuss this point:

“Notably, previous studies have reported brain structural and function changes in PTSD relative to either trauma-exposed controls without PTSD or non-traumatized healthy controls [52-54]. It is difficult to conclude whether the observed alterations were related to disease or traumatic stress. Therefore, for controls, we selected a population that was exposed to the earthquake without developing PTSD to control variables. However, further studies are desirable with an additional group of non-traumatized individuals to provide more comprehensive insight into the functional networks underlying PTSD rather than life stress more generally. ”.

Reviewer 3 Report

The article entitled “Combining deep learning and graph-theoretic brain features to detect posttraumatic stress disorder at the individual level” is well-written and, from my point of view, would be of interest for the readers of Diagnostic. In spite of this and before its publication the following changes should be performed:

  • Lines 124 and 125 speaks about SPM12 altough a link is provided, please give in the text a brief explanation about this software.
  • Line 144 please put the 3 of mm3 as exponent.
  • Line 154 please provide a short explanation about the Gretna toolbox.
  • Support vector machines. Please introduce any reference about it.
  • About references: please check the format. More specifically the following references (numbers 29 and 31) does not seem to be in the format of the journal due to the use of capital letters:
  1. Koch, S.B.J., et al., ABERRANT RESTING-STATE BRAIN ACTIVITY IN POSTTRAUMATIC STRESS DISORDER: A META-ANALYSIS AND SYSTEMATIC REVIEW. Depression and Anxiety, 2016. 33(7): p. 592-605.
  2. Blake, D.D., et al., THE DEVELOPMENT OF A CLINICIAN-ADMINISTERED PTSD SCALE. Journal of Traumatic Stress, 1995. 468 8(1): p. 75-90.

Author Response

Comments to the Author 

The article entitled “Combining deep learning and graph-theoretic brain features to detect posttraumatic stress disorder at the individual level” is well-written and, from my point of view, would be of interest for the readers of Diagnostic.

Authors Reply: We thank the reviewer for these positive comments. 

In spite of this and before its publication the following changes should be performed:

  1. Lines 124 and 125 speaks about SPM12 although a link is provided, please give in the text a brief explanation about this software.

Authors Reply: Thank you for this comment, which we have implemented in the revised manuscript: “Preprocessing of the functional images was carried out using Statistical Parametric Mapping 12 (SPM12) (http://www.fil.ion.ucl.ac.uk/spm), which is a recent and advanced standard neuroimaging processing software.”.

  1. Line 144 please put the 3 of mm3 as exponent.

Authors’ Reply: Thank you for this comment, and we have put the 3 of mm3 as exponent in the revised manuscript.

  1. Line 154 please provide a short explanation about the Gretna toolbox.

Authors’ Reply: Thank you for this comment, which we have implemented in the revised manuscript: “We constructed the network by using the Gretna toolbox (http://www.nitrc.org/projects/gretna/), which allows researchers to perform comprehensive analysis on the topology of brain connectome.”.

  1. Support vector machines. Please introduce any reference about it.

Authors’ Reply: We thank the reviewer for this suggestion. We have introduced a highly cited reference about SVM in Introduction.

  1. About references: please check the format. More specifically the following references (numbers 29 and 31) does not seem to be in the format of the journal due to the use of capital letters:

Koch, S.B.J., et al., ABERRANT RESTING-STATE BRAIN ACTIVITY IN POSTTRAUMATIC STRESS DISORDER: A META-ANALYSIS AND SYSTEMATIC REVIEW. Depression and Anxiety, 2016. 33(7): p. 592-605.

Blake, D.D., et al., THE DEVELOPMENT OF A CLINICIAN-ADMINISTERED PTSD SCALE. Journal of Traumatic Stress, 1995. 468 8(1): p. 75-90..

Authors’ Reply: We thank the reviewer for pointing this out. We have changed the format of these two references in the revised manuscript.